# New Insights into Photobiomodulation of the Vaginal Microbiome—A Critical Review

**DOI:** 10.3390/ijms241713507

**Published:** 2023-08-31

**Authors:** Fernanda P. Santos, Carlota A. Carvalhos, Margarida Figueiredo-Dias

**Affiliations:** 1Faculty of Medicine, Gynecology University Clinic, University of Coimbra, 3000-548 Coimbra, Portugal; carlota.carvalhos@hotmail.com (C.A.C.); marg.fig.dias@gmail.com (M.F.-D.); 2Clinical and Academic Centre of Coimbra, 3004-531 Coimbra, Portugal; 3Gynecology Department, Coimbra Hospital and University Center, 3004-561 Coimbra, Portugal; 4Coimbra Institute for Clinical and Biomedical Research (iCBR), Area of Environment, Genetics and Oncobiology (CIMAGO), Faculty of Medicine, University of Coimbra, 3001-301 Coimbra, Portugal

**Keywords:** lasers, low-level light therapy, therapies, photobiomodulation, microbiota, vagina

## Abstract

The development of new technologies such as sequencing has greatly enhanced our understanding of the human microbiome. The interactions between the human microbiome and the development of several diseases have been the subject of recent research. In-depth knowledge about the vaginal microbiome (VMB) has shown that dysbiosis is closely related to the development of gynecologic and obstetric disorders. To date, the progress in treating or modulating the VMB has lagged far behind research efforts. Photobiomodulation (PBM) uses low levels of light, usually red or near-infrared, to treat a diversity of conditions. Several studies have demonstrated that PBM can control the microbiome and improve the activity of the immune system. In recent years, increasing attention has been paid to the microbiome, mostly to the gut microbiome and its connections with many diseases, such as metabolic disorders, obesity, cardiovascular disorders, autoimmunity, and neurological disorders. The applicability of PBM therapeutics to treat gut dysbiosis has been studied, with promising results. The possible cellular and molecular effects of PBM on the vaginal microbiome constitute a theoretical and promising field that is starting to take its first steps. In this review, we will discuss the potential mechanisms and effects of photobiomodulation in the VMB.

## 1. Introduction

Since the first human genome sequence was published, microbiome studies have grown from culture-based surveys to the sequencing of the whole genomes of organisms belonging to different ecological niches of the human body [1]. Nowadays, it is possible to verify that, through a long evolutionary cohabitation with the human body, this community of microbes (bacteria, archaea, fungi, viruses, and a few protozoan parasites), termed the microbiome, is not only associated with infectious and gastrointestinal diseases, but it has also established a clear role in its host’s physiological functions, such as metabolism, immune development, and behavioral responses [2]. These connections have been the focus of multiple basic and translational studies that continue to expose a compelling range of novel findings, potential mechanisms, and open questions [3].

The National Institutes of Health’s Human Microbiome Project (HMP) was launched in 2007 and was one of the first large-scale initiatives that tried to address a subset of these unanswered questions [1]. This project was divided in two phases; the first sought to determine whether there were common elements to ‘healthy’ microbiomes, in the absence of overt disease. Like other studies, they characterized the specific niche microbial strains, and identified factors that could be responsible for microbiome variations, such as ethnicity, birthing procedures (cesarean vs. vaginal), stress, diet, antibiotic use, exercise, and alcohol consumption [3]. However, one of the main findings was that the host phenotype was predicted by the prevalent microbial molecular function, instead of the entire taxonomic composition of the microbiome alone [3]. This finding supported the second phase of the HMP, which was designed to explore host–microbiome interplay, including immunity, metabolism, and dynamic molecular activity, to gain a holistic view of host–microbe interactions over time. They selected three areas of broad interest to the research community, which were inflammatory bowel diseases; pregnancy and preterm birth (PTB); and stressors that affect individuals with prediabetes [1]. These studies are still underway, but several results have been released.

One of the most studied microbiomes is the gut microbiome, which is recognized as an additional organ because of the interconnection between the human body and its gut microbiome [4,5]. This interplay relies on the microbes’ metabolism, production of neurotransmitters and hormones, and interactions with the immune system, the last of which bring about the association with multiple disease states. The dysregulated gut microbiome, or dysbiosis, is correlated with obesity [6], inflammatory bowel disease [7], irritable bowel disease [8], metabolic syndrome [9], type 2 diabetes [10], cardiovascular disease [11], autism [12], Alzheimer dementia [13], Parkinson’s disease [14], and cancer [4]. Recently, it has also been associated with gynecological disorders, such as bacterial vaginosis (BV), endometrial cancer, polycystic ovary syndrome, endometriosis, and uterine fibroids. Elkafas H. and collaborators describe these interconnections as probably based on estrogen metabolism by the gut microbiome and its ability to activate systemic Th17 inflammation or anti-inflammatory Treg cells [5]. Apart from this connection, the vaginal microbiome (VMB) itself is currently an enormous area of research, based on its impact on women’s health as well as on their newborns’ health [15].

In the HMP, three vaginal sites were considered, specifically, the vaginal introitus, mid-vagina, and posterior fornix. Moreover, there is little distinction between these different locations; they found that *Lactobacillus* sp. dominates in all three regions and that the vagina is a dynamic ecosystem, undergoing a natural variation in the composition of the VMB throughout an individual woman’s life [1].

Several studies have revealed the association between vaginal dysbiosis and increased susceptibility to sexually transmitted infections (STIs) [16] including HIV; infertility; pregnancy complications, including preterm premature rupture of membranes and spontaneous preterm labor (PTB) [17]; and gynecological malignancies [18].

The growing importance of the microbiome in several fields of human health encourages the development of potential therapies that reinstate or maintain healthy microbiota. This has been implemented using prebiotics, probiotics, antibiotics, biofilm disruptors [19], other drug interventions, and fecal [20] or vaginal microbiome transplants [21].

Although, given the paucity of solid data, it is not yet clear which strategies would be the best and there are a range of opportunities for other therapeutic options. It is becoming increasingly apparent that daylight and circadian rhythms play an important role in the development of different diseases [22] and in the success of many treatments, which is why photobiomodulation (PBM) has a huge therapeutic potential. Photobiomodulation is a non-invasive form of phototherapy that utilizes visible and/or near-infrared (NIR) light to trigger a cascade of intracellular reactions. This could treat a multitude of different diseases and disorders, mostly targeting the interactions between the human body and its microbiome [23].

Through this review, we will consider the effects of PMB techniques on the human microbiome, mainly the VMB, and we will discuss the underlying mechanisms.

## 2. The Human Microbiome and Its Diseases

There are many and varied definitions for the microbiome; one of the most universally accepted was given by Lederberg and McCray in which the microbiome is defined as a group of microorganisms within an ecological environment, space, or body, and that lives in a close physical association of mutualism or commensalism [24,25].

This group of microbes (bacteria, archaea, fungi, viruses, and a few protozoan parasites) that comprise the microbiome was the subject of a substantial amount of research in the last decade [26], particularly in its interaction with the human body [23].

These microorganisms coexist in symbiosis with our body and have a major impact on almost every organ system, including the oropharynx, gut, skin, and genital tract. They influence digestion, immunity, cognitive functions, and even longevity. The human microbiome has an overwhelming influence on the maintenance of health in general and modulates individual susceptibility to many diseases [24,27].

The accumulation of numerous scientific data and knowledge related to the microbiome has produced a paradigm shift in understanding health and disease [24]. In fact, given its established importance, the microbiome is nowadays recognized as “our last organ” [28].

The gut microbiome is the most widely studied microbiome. It is known that the gut microbiome interacts with the brain microbiome in what is known as the “gut–brain axis” [27]. Bidirectional gut–brain interactions regulate key physiological and homeostatic functions, including food intake, immune regulation, and sleep. The brain connectome, gut connectome, and gut microbiome make up the three nodes in the gut–brain microbiome network. Communication within this system is bidirectional, with multiple feedback loops, and involves interactions between different channels [29]. The gut microbiota can communicate with the brain either directly, via different signaling molecules, or indirectly, via the gut–brain axis through microbe-derived neuroactive molecules. Similarly, the brain can modulate the microbiome either directly or via modifications of the gut microbial environment using gut-derived neuronal, immune, or neuroendocrine molecules [30,31].

This axis is essential for understanding numerous diseases, such as irritable bowel syndrome [7] and functional dyspepsia, but it also may play a key role in the understanding of the pathophysiology of several digestive, psychiatric, and neurological disorders [30,31].

Similar to the gut–brain axis, the gut–vagina axis has been argued to have an essential role in female health and reproductive outcomes (conception and birth) [31,32]. Several studies have already confirmed that the gut and genital tract microbiota of females represent very complex biological ecosystems that are in continuous communication with each other. Some evidence of this connection came from: the identification of common bacteria phyla, such as *Firmicutes*, *Bacteroidetes*, *Proteobacteria*, *Actinobacteria*, and *Fusobacteria* [32]; the recognition that vaginal *Lactobacillus* sp. originates from the gut [32]; the finding that the primary source of Group B *Streptococcus* vaginal infections in pregnant women is the intestinal tract [32]; the study of the regulation of systemic estrogens levels through *secretion of β-glucuronidase* by gut microbiota [33]; and studies that demonstrate that a dysbiotic vaginal microbiota (as in BV) can stimulate a similar phenotype in the gut [34,35]. Through mouse models, it was possible to demonstrate that *G. vaginalis* infection suppresses the anti-inflammatory cytokine IL-10 and stimulates activation of NF-kB, TNF-α expression, and myeloperoxidase activity in both the vagina and colon. It also increases the population of *Firmicutes* and *Proteobacteria* [major producers of lipopolysaccharide (LPS)], reduces *Bacteroidetes* in the vagina, and stimulates gut microbiota LPS production resulting in dysbiosis characterized by increased *Proteobacteria/Bacteroidetes* and *Firmicutes/Bacteroidetes* ratios [32].

Usually, an increase in the *Firmicutes/Bacteroidetes* ratio results in a decrease in the total amount of short-chain fatty acids (SCFAs) [34]. SCFAs, such as butyrate, acetate, and propionate, are potent signaling molecules produced by the microbiome. They maintain intestinal homeostasis through maintenance of intestinal epithelial integrity and prevention of bacteria and LPS leakage into the systemic circulation (leaky gut) [35]. Hence, the reduction in SCFAs is permissive to LPS-induced inflammation and increased risk of metabolic disorders, including obesity and type 2 diabetes mellitus. Thus, *G. vaginalis* infection could stimulate gut inflammation (e.g., colitis) and induce a systemic proinflammatory state [33]. The homeostatic and immunomodulatory effects of SCFAs in the gut are better understood compared to those of the vagina, and, despite their origin, SCFAs in the gut have also been described as having obstetric implications [36]. In this way, elevated gut SCFAs during pregnancy may remotely reduce the risk of infection and inflammation associated with spontaneous PTB [36]. As SCFAs prevent leaky gut, this reduces the eventual hematogenous spread of bacteria and LPS to the uterus, placenta, or amniotic cavity, thereby preventing the production of LPS-induced inflammatory mediators and prostaglandins that trigger labor [32]. The link between dysbiotic gut microbiota and the risk of spontaneous PTB [37] requires further investigation, as this could explain why treating vaginal infections in some women does not reduce the risk of preterm delivery [32]. Gut dysbiosis has been shown to relate to other obstetric complications, such as gestational diabetes, preeclampsia, and other birth complications [37].

### 2.1. Vaginal Microbiome (VMB)

The VMB is a dynamic, sensitive microenvironment that changes in response to medications, sexual activity, contraceptive use, hormonal changes (pregnancy, menstrual cycle), and external factors such as diet, smoking, or stress [38].

In women, the VMB is often dominated by *Lactobacillus* sp.; its production of lactic acid as a metabolic bioproduct lowers the vaginal pH and inhibits the overgrowth of potentially pathogenic microorganisms, thus limiting diversity [5].

The VMB is a dynamic ecosystem that varies between women, depending on several factors. Ravel et al., in 2011, introduced the concept of community state types (CSTs) after discovering that the microbiomes of women could be clustered into five core community groups. Four of these groups (CST I, II, III, and V) are dominated by *Lactobacillus* sp. (CST I—*L. crispatus*; CST II—*L. gasseri*; CST III—*L. iners*; CSTV—*L. jensenii*), while the fourth group (CST IV) differs by having a relatively low abundance of *Lactobacillus* and higher proportions of anaerobic bacteria [39].

Since then, studies have further developed this classification system and added subtypes to the five CSTs, most of which distinguish between variations in CST IV. An example of a CST IV subtype is the one dominated by *Bifidobacterium*, a group of Gram-positive anaerobes that are known to colonize the vagina, oral cavity, and gastrointestinal tract. These species can produce lactic acid and can tolerate a low pH, the latter of which is typical of healthy vaginal fluid. This suggests that *Bifidobacterium* sp. may be as protective as *Lactobacillus* sp. in preventing vaginal colonization by pathogenic organisms [38].

Meanwhile, this classification system approach simplifies the VMB composition, and it continues to be an important foundation for its research and understanding [40].

### 2.2. The Impact of Vaginal Dysbiosis on Health

Vaginal microbiome composition, and particularly the abundance of *Lactobacillus*, is increasingly described as having a central role in the regulation of female reproductive tract inflammation [16]. However, the protective *Lactobacillus* community can be easily disrupted, resulting in vaginal dysbiosis [38]. Women with a lower proportion or quantity of *Lactobacillus* and a higher proportion or abundance of anaerobes are predisposed to a variety of adverse vaginal health conditions [38].

Vaginal infections can occur when vaginal dysbiosis allows the overgrowth of opportunistic organisms such as *E. coli*, *G. vaginalis*, and *Bacterial-vaginosis-associated bacteria*, or when exposed to a range of pathogenic organisms such as *C. trachomatis* or *N. gonorrhoeae.* Prompt detection and treatment of these infections are crucial, as they can predispose women to a range of reproductive health conditions, such as pelvic inflammatory disease, infertility, and PTB [38].

Consequently, the VMB has an essential role in the protection against the acquisition of STIs such as HIV, HSV, HPV, gonorrhea, *chlamydia*, *trichomonas*, and syphilis. *Lactobacillus* allows the maintenance of a low-inflammatory environment and can limit the acquisition and transmission of STIs through different mechanisms, such as lactic acid/bacteriocin production and adhesion to cells/pathogens. Neutrophils are key modulators of inflammation, and their interaction with the microbiota impacts the immune response to STIs. However, the mechanisms of action are still inadequately understood [16].

The vaginal microbiome also affects the incidence of urinary tract infections, which are more common among women with vaginal *E. coli* colonization and with a low prevalence of *Lactobacillus* [40,41,42].

In dysbiosis, the higher growth of several BV-associated bacteria was associated with an increased risk for pelvic inflammatory disease [40,43] and endometritis [40,44].

The VMB is currently the most studied microbiome in the human body concerning pregnancy and reproductive health.

Generally, a low-diversity, *Lactobacillus*-dominated vaginal microbiome is considered the healthiest [45]. During a healthy pregnancy, the *Lactobacillus* dominance becomes even more pronounced, likely due to the large increase in estrogens [46]. In the postpartum stage, when estrogen and progesterone decline and lochia is flowing, the VMB has been described as being more diverse than during pregnancy, with lower proportions of *Lactobacilli* [47,48].

Vaginal dysbiosis has also been associated with poor fertility treatment outcomes, miscarriage, PTB, preeclampsia, gestational diabetes, and excessive gestational weight gain, which are all common gestational complications [49,50,51,52]. There is also strong evidence suggesting that maternal and early infant microbiomes could be associated with neonatal outcomes, such as fetal growth, metabolic acidosis, early metabolic or endocrine disturbances, and outcome of neonatal intensive care [51].

If these negative correlations are corroborated, patients could benefit from screening and treatment to achieve successful fertility treatment [53] and pregnancy outcomes [54,55,56,57,58,59].

## 3. Photobiomodulation: Definition and Mechanism of Action

In 1903, Finsen earned the Nobel Prize in Medicine and Physiology for his work in treating cutaneous tuberculosis with UV light and smallpox with red light [3]. Since then, therapy through light has fallen from favor. In 1967, Endre Mester and colleagues rediscovered laser light as a treatment modality [60]. Nowadays, he is considered the “father of photobiomodulation” because he tried to destroy malignant tumors, implanted in the skin of rats, through use of a ruby laser (694 nm), and despite failing to achieve the cure, he found that through this low-power laser light, it was possible to induce a positive effect on wound healing and hair regrowth in the skin of exposed rats. This effect was called “laser biostimulation” [61].

A consensus nomenclature of PBM therapy was defined by Anders et al. as “a form of light therapy that utilizes non-ionizing light sources, including lasers, light emitting diodes (LEDs], and broadband light, in the visible and infrared spectrum” [62,63] (Figure 1). These therapeutic effects are achieved at a window delineated between approximately 600 and 1200 nm [64] (Figure 1).

A wide spectrum of studies has been published focusing on the various physiological effects of visible red light and near-infrared radiation. Distinct light parameters must be considered to evaluate different physiological effects and their effectiveness. It is important to evaluate the impulse type, the dose, the time of radiation, and the wavelength [65].

The wavelength determines the depth of tissue penetration [66] (Figure 2). Therefore, shorter wavelengths (600 to 700 nm) are the best for treating superficial diseases, whereas those at longer wavelengths (780 to 950 nm) are preferred for diseases affecting deeper tissues. However, at longer wavelengths, water becomes an important absorber, which means that penetration decreases [64].

Some authors describe a “biphasic dose response curve”, in which lower dosages will stimulate the biological process, and higher doses could have a negative therapeutic effect [65,68]. Recently, Amaroli A. and colleagues reported window effects when mitochondria were irradiated with diode laser light of 980 nm, suggesting that mitochondria effects are not biphasic, but instead seem to occur in narrow windows of positive effect/no effect/negative effect. They also reconsidered the light parameters able to interact with cell photoacceptors [69,70]. In addition to energy’s penetration into biological tissues at the site with the lowest losses, it acts locally (and possibly systemically) to affect cellular metabolism, cellular signaling, inflammatory processes, and growth-factor production [64,65,71]. Although the details of these mechanisms are still under investigation, it is possible to describe some of them, which has allowed the wide implementation of PBM in different fields of medical care.

It has been shown that many cellular molecules are able to absorb several wavelengths of light. Examples include hemoglobin, melanin, and porphyrin; these molecules are not influenced by the photobiomodulation effects; instead, this is due to the absorption of incident photons by photoacceptors, also called endogenous chromophores [64]. Evidence suggests that the primary cellular photoacceptors are the copper centers of cytochrome c oxidase (CCO), a complex protein that functions as unit IV of the mitochondrial respiratory chain. It has been suggested that tissues with higher numbers of mitochondria respond better to PBM than do those with lower numbers [64]. When CCO absorbs red and near-infrared light, the nitric oxide (NO) dissociates, reactive oxygen species are produced, the mitochondrial membrane potential increases, intracellular ATP and cyclic AMP increase, and changes in Ca2+ concentration and numerous downstream effects occur, such as transcription factor activation (NF-kB, HIF-1. and RANKL) [72] (Figure 3).

These signaling molecules could also change the expression of a multitude of gene products, including structural proteins, enzymes, and mediators of cell differentiation, division, migration, and apoptosis [23] (Figure 3). In addition to CCO, other suggested photoacceptors include opsins, flavins, flavoproteins, and porphyrins [63,68].

Sommer A. demonstrated that mitochondrial ATP synthesis can be triggered by various wavelengths of light, pointing out that CCO could not explain all photobiomodulation effects [73]. Wang et al. also found out that blue (420 nm) and green (540 nm) light enhanced the differentiation of human adipose-derived stem cells, highlighting the photomodulation effect, but with confusion as to the mechanisms that lead to ATP upregulation [74].

By Sommer’s description, upregulation of mitochondrial ATP results from a physical process involving a reduction in interfacial water layers’ viscosity in the irradiated mitochondria [73]. Other authors have also published mechanisms of PBM beyond the CCO, such as direct and indirect changes to oxidative stress, activation of several ATP-dependent channels, and production of NO, as well as through photophysical processes and interactions with the cytoskeleton and other cell proteins [72,75].

Photobiomodulation therapies have been applied in the treatment of a multitude of diseases [26]. It has been applied to several painful conditions [76], at wound healing [77], preventive treatment for acute radiotherapy dermatitis [78], as a treatment for oral mucositis [78], and recently as a potential radiosensitizing agent for human cervical cancer cells [79,80].

Several studies are underway; exactly 109 clinical studies are being carried out, according to the ClinicalTrials.gov database, accessed on the 28 July 2023. At the same time, the research on microbiome and light treatment is limited.

### 3.1. Photobiomodulation of the Microbiome

The area of “photobiomics” is a very new field of science; therefore, the discussions of possible mechanisms of action are mainly theoretical.

PBM can induce local effects, but it can also induce significant systemic effects, as argued by its effect on cell metabolites (metabolome), which are crucial in determining signaling profiles, modifying phenotypic expression (such as the ability to switch macrophage phenotype from M1 to M2) and post-translational modification of proteins by tyrosination, methylation, or SUMOylation [81]. Another interesting mechanism is related to the potential interaction with the human microbiome, justifying its applicability for several systemic diseases [68,82,83,84,85,86,87].

PMB therapeutics have been particularly investigated at the microbiome/gut/brain axis [27,29]. Preclinical evidence has firmly established bidirectional interactions among these major players. Liebert et al. published a study finding that NIR wavelengths delivered to the abdomen of healthy mice can produce a significant change in the gut microbiome, inducing a high level of bacteria associated with a healthy microbiome [26]. Bicknell B. et al. evaluated the effect of infrared laser treatment (904 nm; 700 Hz pulse frequency, 861.3 total joules) delivered to the abdomen of breast cancer patients, three times per week for 11 weeks, and identified an increase in the number of beneficial gut bacteria (*Akkermansia*, *Faecalibacterium*, and *Roseburia*) and decrease in the pathogenic correlative [88].

Research has also shown that gut microbiota can signal to the nervous system through the gut-associated lymphoid tissue via immune responses, redox signaling, the endocrine system, and the enteric/vagus nerve pathway [30]. Several animal studies have evaluated these mechanisms and suggest that there are interactions between gut dysbiosis and the development of a multitude of pathological conditions, as mentioned previously [30]. Nonetheless, the mechanisms of photobiomodulation interaction are still uncertain. Further research must elucidate if the light is primarily absorbed by the microbial cells belonging to the human microbiome, by the host cells that surround it (or indeed cells that are distant from them), or by a combination of both microbes and host cells. Most studies have evaluated the effects of PBM on photoreceptors of mammalian cells. However, a Russian team demonstrated that several species of Gram-positive and Gram-negative bacteria and fungal (including yeast) cells do respond to PBM [89].

Zanotta et al. demonstrated that PBM (λ 970 nm, power 2.5 W, irradiance 200 mW/cm^2^, fluence 6 J/cm^2^, time 30 s, continuous wave) is effective in reducing the inflammatory burden in cancer patients affected by oral mucositis. They also reported a positive influence in the composition of the oral microbiome in patients who improved (by reduction of, e.g., *Neisseria* and *Haemophilus*) [90].

Since 2021, the Mucositis Study Group of the Multinational Association of Supportive Care in Cancer/International Society for Oral Oncology, has recommended PBM therapy for oral mucositis prevention in patients treated with chemotherapy [91].

The literature shows that visible and near-infrared light affects the bacteria, through direct and indirect effects, independently of Gram-negative or Gram-positive bacteria [92].

As mentioned previously, the direct interaction is based on the primary effect of light on “photoacceptive” molecules. Indirectly, pigments can generate these PBM effects. The pigmented bacteria can be killed by light at a low-level dose reliable for PBM, and non-pigmented bacteria associated with the colony can be involved in this lethal effect [92].

The microbiome could also be changed due to interaction with inflammatory responses [23]. PBM can increase salivary levels of interleukin-1 receptor antagonist and IL-10, as well as total antioxidant capacity, changing salivary parameters that are essential for avoiding oral microbiota dysbiosis [92].

Balle C. et al. identified a relationship between the oral and vaginal microbiota of South African adolescents, with a high prevalence of bacterial vaginosis. The salivary microbiota of BV participants was significantly more diverse than in those with *lactobacillus*-dominated communities. With this study they highlighted the strong epidemiological evidence that exists between BV, periodontal disease and PTB [93].

These interconnections suggest the consideration that PBM could also have applicability for the VMB.

### 3.2. Potential Applications for the Vaginal Microbiome (VMB)

The applicability of PBM to the vaginal microbiome is promising, but still under-researched. However, potential targets can be discussed (Table 1).

One of the main mechanisms of PBM is its ability to induce photo-dissociation of NO from heme proteins, leading to energy metabolism modulation by reducing oxygen consumption (Figure 3). NO is a central signaling molecule of biological systems that, in addition to involvement in vasodilation, endothelial function, anti-aggregation of platelets, nerve transmission, host defenses, and cellular energy, also functions as the main component of the vaginal immune response to BV [31]. It was also observed that alterations in commensal bacteria trigger the release of hsp70 into the extracellular matrix, which subsequently induces an increase in NO levels. Afterwards, NO stimulates the production of anti-inflammatory cytokines that prevent the invasion of pathogenic bacteria [31]. Studies have also been published suggesting that *Lactobacillus* sp. could also produce NO from exogenous arginine [94,95]. Therefore, disruptions of the NO balance could lead to dysbiosis. In this context, studies have been carried out aiming to identify strategies to deliver drugs targeting NO. Based on its ability to augment NO bioavailability, PBM could be a potential option. It will be interesting to investigate the exact parameters of light that could induce biostimulation or bioinhibition, since NO is known to have a dual role; for example, NO inhibits HIV1 replication in acutely infected cells, whereas overproduction of NO stimulates HIV-1 reactivation in chronically infected cells [96].

At the gut-vagina axis, several therapeutic options have been studied to restore gut and vagina homeostasis, but although these studies have been promising, the question requires further investigation [19,20,21]. In this regard, biomodulation through light could have applicability, since treatment with PBM has been shown to result in changes in the human microbiome by increasing *Akkermansia muciniphila*, *Bifidobacterium* sp., and *Faecalibacterium* sp., all recognized as being correlated with a healthy microbiome, and decreasing the *Firmicutes/Bacteroides* ratio [26]. These gut–vagina interactions clearly demonstrate that dysbiosis in any of the biomes requires a holistic intervention, particularly for recurrence of disorders. Therefore, the applicability of PBM for BV conditions could have a huge medical impact, based on the association of this dysbiosis with STI acquisition, infertility, miscarriage, or PTB [32].

Marti H. et al. studied the applicability of PMB to *Chlamydia*, a common sexually transmitted bacteria related to multiple female sequelae, such as pelvic inflammatory disease, ectopic pregnancy, and infertility. They found that water-filtered infrared A, a short-wavelength infrared radiation with a spectrum ranging from 780 to 1400 nm, could significantly reduce infectivity of extracellular infectious elementary bodies in two different cell lines (Vero, HeLa), without inducing cytotoxicity within host cells. Thus, this irradiation might be a promising approach to this infection [97].

Vulvovaginal candidiasis is the second-most-common vaginitis after BV, and it is estimated that 30–50% of women will suffer from it at least once in their lifetime [98]. It is characterized by acute vaginal inflammation due to the overgrowth of normal commensal *Candida* sp., mainly *Candida albicans* [99]. How diverse *Candida* sp. interact with host cells has rarely been addressed. Pekmezovic M. et al. used a time course infection model of vaginal epithelial cells and dual RNA sequencing and showed that different *Candida* sp. exhibit distinct pathogenicity patterns, which are defined by highly species-specific transcriptional profiles during infection of vaginal epithelial cells [99]. Moreover, energy production and mitochondria are central hubs in innate immunity. Through this research, it was possible to verify that, at early stages, the host cells exhibit a homogeneous response to all species, which was characterized by sublethal mitochondrial signaling inducing a protective type I interferon response. These data illustrate that type I IFN increases epithelial resistance to *Candida* infection through the induction of pro-inflammatory responses. At the later stages, the host response diverges and is damage-driven [99]. The characterization of mechanisms of action can lead to potential therapeutic interventions. Therefore, based on these explanations, PBMt could be a therapeutic option. This has been shown experimentally mainly through anti-fungal blue light therapies. In vitro studies have demonstrated that 415 nm blue light can have effective anti-fungal results with low levels of damage to surrounding epithelial cells [100]. The proposed mechanism underlying these bioinhibitive effects was the oxidative stress caused by the accumulation of ROS, as induced by the light, although it could have also resulted from stimulation of CCO at the mitochondrial membrane [100].

**Table 1 ijms-24-13507-t001:** Potential interactions: the vaginal microbiome and photobiomodulation.

Vaginal Microbiome Component	PBM Mechanism	Reference
NO, the main component of the vaginal immune response to bacterial vaginosis. *Lactobacillus* sp. could also produce NO from exogenous arginine. Disruptions of NO balance could lead to dysbiosis.	PBM augments NO bioavailability. NO stimulates the production of anti-inflammatory cytokines that prevent the invasion of pathogenic bacteria.	[94,95,96]
Gut–vagina axis	PBM induces changes in the human microbiome by increasing *Akkermansia muciniphila*, *Bifidobacterium* sp., and *Faecalibacterium* sp. (correlated with healthy microbiome) and decreasing the *Firmicutes/Bacteroides* ratio.	[26,32]
*Chlamydia* sp.	Water-filtered infrared A (λ 780 to 1400 nm), can significantly reduce infectivity of extracellular infectious elementary bodies in HeLa cell lines.	[97]
*Candida* sp.	415 nm blue light had anti-fungal effects.	[99,100]

## 4. Conclusions

The human microbiome confers metabolic capabilities exceeding those of the host organism alone, making it an active participant in the host’s physiology. The recognized ability of light’s interference with microbiomes, specifically through PBM, opens a field of promising therapeutic interventions. Red/NIR light can interact with cells, leading to changes at the molecular, cellular, and tissue levels.

PMB therapeutics have been particularly investigated at the level of the gut microbiome, but the underlying mechanisms and interconnections make possible a physiological applicability to VMB. This interplay could be related to the nitric oxide effect, since PBM can augment the bioavailability of NO, which is essential to vaginal immunity. It was also determined in published work that there are potential benefits to the use of light in the treatment of *Chlamydia* sp. and *Candida* sp. infections.

Further studies are required to fully elucidate the light’s parameters, real effects, and underlying mechanisms.

## Figures and Tables

**Figure 1 ijms-24-13507-f001:**
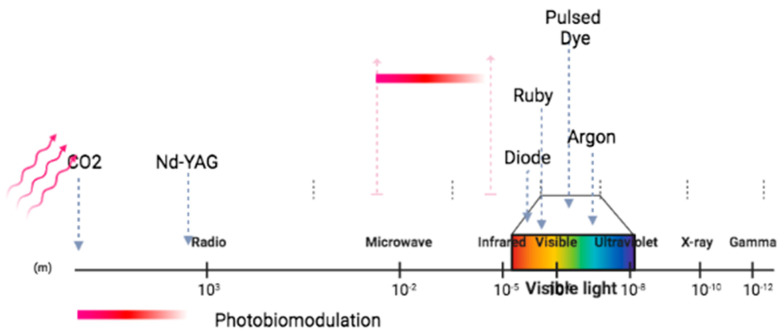
Schematic representation of laser light [64]. Red arrows demonstrate photobiomodulation wavelengths. Gray arrows demonstrate different examples of laser light. Created with BioRender.com.

**Figure 2 ijms-24-13507-f002:**
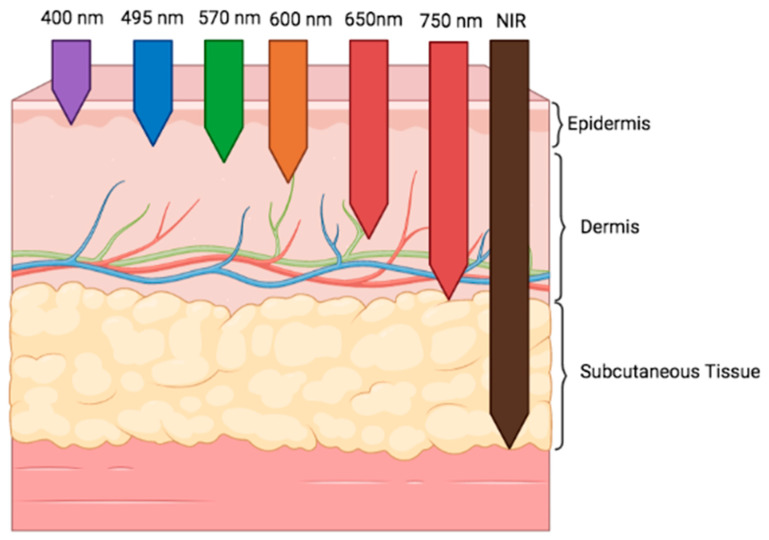
Depth of light penetration into skin by wavelength [66,67]. NIR: Near infra-red. Created with BioRender.com.

**Figure 3 ijms-24-13507-f003:**
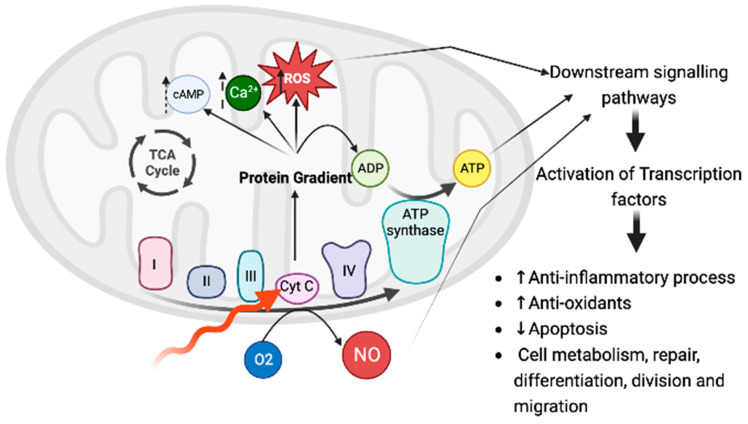
Representation of a potential mechanism of PBM’s effect. The application of red and near-infrared light could be absorbed by the enzyme cytochrome c oxidase, which is in unit IV of the respiratory chain in the mitochondria. Effects of photon absorption include increases in ATP, a brief burst of reactive oxygen species generation, an increase in nitric oxide, modulation of calcium levels, and activation of a wide range of transcription factors leading to improved cell survival, increased proliferation and migration, and new protein synthesis [26,68,72]. Created with BioRender.com.

## Data Availability

No new data were created in this study. All the data reported in this review were found in original articles cited in the text.

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
