# Peer review of "New Insights into Photobiomodulation of the Vaginal Microbiome—A Critical Review"

_ijms, 2023, doi:10.3390/ijms241713507_

Round 1

Reviewer 1 Report

paper is not focused, authors need to concentrate on Vaginal Microbiome only not the gut or other parts of the body, then expand more on PBM, if possible use a systematic review approch

abstract, and the summary of wavelengths used for modulating the Vaginal Microbiome and power 

Section 3.1, considering the similarities between oral and vaginal soft tissues, you need to add more on orofacial PBM applications, line 373-375, you need to add more on orofacial uses, including the use in reducing orthodontic pain, or acceleration of orthodontic tooth movement (Lasers Med Sci. 2014;29(2):559-64. ; Borzabadi-Farahani A, Cronshaw M (2017) Lasers in orthodontics. In: Coluzzi DJ, Parker SPA (eds) Lasers in dentistry—current concepts. Springer International Publishing, Cham, pp 247–271. https://doi.org/10.1007/978-3-319-51944-9_12)

Line 381-386, add proliferation and migration of dental stem (J Photochem Photobiol B. 2016;162:577-582.)

result section, add a table and include all the relevant trials on the use of PBM for Vaginal Microbiome modulation

conclusion is too long, revise, add the main findings

needs revision

Author Response

The manuscript was revised by MDPI English editing.

We included articles that conjugate PBM treatment and oral microbiome.

All suggestions were very useful, and corrected

Reviewer 2 Report

Dear colleagues,

Through a Critical Review, an attempt has been made to highlight the potential therapeutic aspect of photobiomodulation in the gynaecological field, particularly concerning the vaginal microbiome. The topic is timely within the realm of photobiomodulation, as underscored by recent and insightful reviews.

doi: 10.1007/s10103-018-2594-6

doi: 10.3390/ijms23031372

doi: 10.1089/photob.2019.4628

The major concern of this review is that upon completing the reading, the impact of photobiomodulation for innovative therapies on the vaginal microbiome remains unclear.

Primary doubts and considerations include:

The central issue lies in the fact that while the gynaecological portion is presented with clarity and expertise, the sections pertaining to laser applications consist mainly of partially expressed well-known concepts requiring deeper exploration. Above all, there is an absence of a section connecting the vaginal microbiome with photobiomodulation that would support the review's objectives. It is recommended to entirely eliminate section 3.1 "Applications to human diseases." The review is already quite extensive, and the therapeutic application of photobiomodulation across various medical domains has been extensively discussed in numerous existing literature reviews. Instead, the focus should shift towards interactions between photobiomodulation and the microbiome in general, with specific emphasis on the vaginal microbiome.

Minor Considerations:

1) "Abstract: Photobiomodulation (PBM) uses low levels of visible light, usually red or near-infrared, to treat a diversity of conditions."

"Near-infrared" is not visible light; please rephrase the sentence accordingly.

2) "Keywords"

Keywords should be updated to align with the terms indicated by MeSH (Medical Subject Headings) – link: https://www.ncbi.nlm.nih.gov/mesh

3) "Photobiomodulation is an umbrella term to describe the therapeutic application of low levels of red and/or near-infrared (NIR) light."

It is suggested to reference visible light in general (not only red). For instance, blue light is also employed.

4) Genus and species MUST be written in italics!

5) "The biphasic dose responses, which could concurrently be through biostimulation or through bioinhibition effects."

"The dose follows a 'biphasic dose-response curve,' adhering to the Arndt–Schulz law. Lower dosages stimulate biological processes, while higher dosages yield a negative therapeutic effect."

While many texts describe biphasic response definitions or hormetic effects, there is no unequivocal evidence of such effects. In some cases, dose-related window effects are observed. On the other hand, literature on electromagnetic fields of intensities and frequencies different from light often reports window effects. In a special issue published in Oxidative Medicine and Cellular Longevity, for example, authors reported window effects when mitochondria were irradiated with 980nm light.

doi: 10.1155/2021/6626286

6) "For example, it has been found that PBM can produce ROS in normal cells, but when used in oxidatively stressed cells or in animal models of disease, ROS levels are lowered."

This example does not exemplify the previous statement. In this case, two distinct metabolic conditions are involved, not biphasic effects.

7) "Evidence suggests that the primary cellular photoacceptors are the copper centers of cytochrome c oxidase (CCO), a complex protein that functions as unit IV of the mitochondrial respiratory chain. It has been suggested that tissues with higher numbers of mitochondria respond better to PBM than those with lower numbers."

While CCO is a cornerstone of photobiomodulation, recent reviews have challenged its centrality, despite its important role.

doi: 10.21037/atm.2019.01.43

Although responding to one dogma with another is not advisable, it is evident that photobiomodulation is a process far more intricate than commonly synthesized. In a review aiming to connect the realms of prokaryotes, eukaryotes, and light, the manifold actions of photobiomodulation should be better described and interpreted. I recommend reading the section "4.1. Photobiomodulation and Mitochondrial Bioenergetics" in:

doi: 10.3390/ijms22094347

and the article doi: 10.3390/ijms23031372.

8) "As mentioned before, besides local application, significant systemic effects could be produced. Some related mechanisms are the effect on cell metabolites (metabolome), which is crucial in determining signaling profiles, modifying phenotypic expression (such as the ability to switch macrophage phenotype from M1 to M2), and post-translational modification of proteins by tyrosination, methylation, or SUMOylation (82). Finally, the effects of PMB on the human microbiome could also explain its applicability for several systemic diseases (83–89)."

The utility of these sentences in the work remains uncertain. If the effect of photobiomodulation on the immune system can impact the review's objective, it might be worthwhile to dedicate an explanatory paragraph to it, along with sections on the microbiome in general and, specifically, the vaginal microbiome.

9) “3.2. Potential applications to the vaginal microbiome (VMB)”

Unfortunately, this is the weakest part of the entire work. I hope that following the recommended readings and revisions, this critical aspect of the study can be strengthened by highlighting the effects of photobiomodulation on the vaginal microbiome (if relevant research exists). If such studies are absent, it would be prudent to speculate and better correlate the preceding paragraphs.

Best regards

Author Response

All suggestions were taken into account

Round 2

Reviewer 1 Report

the paper does not add new information to the literature, you need an article in systematic review form, you need define your PICO, device search terms, and select the right papers and report on their deficiencies or findings

needs improvement

Reviewer 2 Report

Dear colleagues, 

I would like to express my gratitude for the thorough revision work that has enhanced your manuscript. I have no further requests at this time.

Congratulations and best regards,

Sincerely,

Author Response

Thank you very much for all comments and suggestions. 

Sincerely, Fernanda Santos